# Hydrogeochemical Evolution of an Aquifer Regulated by Pyrite Oxidation and Organic Sediments

Eugenio Sanz [1], Catalina Bezares [1], Carlos Pascual [2], Ignacio Menéndez Pidal [1,*] and Cristina Fonolla [1]

[1] Departamento de Ingeniería y Morfología del Terreno, Escuela Técnica Superior de Ingenieros de Caminos, Canales y Puertos, Universidad Politécnica de Madrid, 28040 Madrid, Spain; eugenio.sanz@upm.es (E.S.); c.bezares@alumnos.upm.es (C.B.); cristina.fonolla@upm.es (C.F.)

[2] C/Real 65, 42002 Soria, Spain; capascual-1@telefonica.net

\* Correspondence: ignacio.menendezpidal@upm.es

**Abstract:** Detailed full-scale groundwater monitoring was carried out over a period of nine years, sampling at selected points along the groundwater flow direction in its final stretch. This established the hydrogeochemical evolution along the flow of a natural system formed by a calcareous aquifer which discharges and then passes through a quaternary aquifer of lake origin which is rich in organic matter. This evolution is highly conditioned by the oxidation of pyrites that are abundant in both aquifers. In the first aquifer, one kilometre before the discharge location, oxidizing groundwater crosses a pyrite mineralization zone whose oxidation produces an important increase in sulphates and water denitrification over a short period of time. In the quaternary aquifer with peat sediments and pyrites, water experiences, over a small 500 m passage and residence time of between three and nine years, a complete reduction by way of pyrite oxidation, and a consequent increase in sulphates and the generation of hydrogen sulphuric acid. This is an example of an exceptional natural hydrogeological environment which provides guidance on hydrogeochemical processes such as denitrification.

**Keywords:** redox processes; groundwater; nitrate attenuation; pyrite oxidation; peat sediments

## 1. Introduction and Objectives

Understanding the spatial distribution of redox processes in aquifers and quantifying the rate of reactions is essential for assessing groundwater quality because the fate of many pollutants largely depend on these processes and on the self-purifying effects of aquifers [1,2]. Many of these processes are conditioned by microbial metabolism, whose energy for maintenance and growth depends on the oxidation of organic or inorganic species (e.g., pyrite). Organic carbon is the most common electron donor in groundwater, as is the case of peat [3]. Microorganisms first oxidize organic carbon by using dissolved $O_2$ as an electron acceptor through aerobic respiration because it is the most energetically favourable reaction. When the dissolved $O_2$ is consumed, anaerobic practitioners begin to use nitrate ($NO_3^-$) as an electron acceptor during denitrification, increase in sulphates, and other inorganic redox systems.

Nitrate reduction by the oxidation of organic matter (denitrification) from soils was well documented by [4]. Degraded peat soils in artificial wetlands showed potential to serve as a substrate for the cleanup of nitrate-laden agricultural runoff (degraded peat soils in constructed wetlands) [5]. With organic materials and through the Fe (III)/Fe (II) cycle, nitrate reduction in groundwater can occur [6].

Iron sulphide was identified as a source of electrons in groundwater systems [7–9]. The oxidation of pyrite in aerobic environments by microorganisms is quite well-researched, but there are fewer studies in anoxic environments. Oxidation can lead to the degradation of groundwater [10], as occurs with acidic mine water [11]. However, pyrite is also used to denitrify groundwater [12].

Thus, pilot tests are carried out in artificial wetlands where pyrite is used to denitrify polluted water [13]. Similar denitrification processes are produced using natural organic and pyrite-rich materials from geological formations that contain them [14]. The denitrification process was also evaluated in natural aquifers with pyrite and where the residence time of groundwater is long [2,15].

In several of these aquifers, as oxidized groundwater moves into aquifer sectors containing iron sulphide, their oxidative dissolution increases sulphate concentrations according to the age of the groundwater in the aquifers [7]. In recent decades, stable sulphur isotope analysis (SSIA) has been established as a powerful tool for tracking pyrite oxidation and other sulphur cycling processes in various environments [16–20].

On the other hand, knowledge of the age gradients of groundwater (for example, by using tracers) along a flow line in shallow aquifers allows knowledge of the reduction rates of $O_2$ and nitrates [21–23]. The rate at which $O_2$ is reduced significantly influences the susceptibility of aquifers to many pollutants and the vulnerability of sensitive polluting aquifers to redox processes.

This study investigates the case of natural peatlands which are the discharge zone of limestone aquifers where pyrites are abundant in a dispersed and generalized way, a circumstance that confers a singular hydrogeological environment. Indeed, it is a unique case since the association of peat bogs with extensive pyritised areas in natural conditions, such as occurs here, does not occur in other parts of the world; hence, its study is important.

A significant portion of the recharge surface of these calcareous aquifers is dedicated to agriculture, and there are also pig farms, which are sources of nitrate contamination. This hydrogeological device, i.e., limestone aquifers with pyrites and peatlands in discharge areas and in downstream alluvial areas, is exceptional and there is no other known area such as this. Its study is an opportunity to determine how the natural processes of reduction and oxidation (denitrification, for example) operate under natural conditions in groundwater, and can serve to preserve a similar area (Añavieja) and perhaps contribute ideas for the construction of artificial wetlands. This case can be used as a hydrogeochemical model of how the self-purifying effect of a natural aquifer with pyrites and organic sediments functions. Although this hydrogeological environment is found in two nearby sites, the Añavieja and Ágreda aquifers, only in the latter do we have enough data to tackle its study. The Ágreda aquifers form a set of two perfectly interconnected systems: it is an extensive carbonate aquifer with pyrites that subterraneously feeds another small and shallow Quaternary aquifer, where organic materials predominate. In both, there was the opportunity to know and compare the chemistry of the water that circulates and flows out from the first aquifer (springs known as Los Ojillos del Keyles), which, in turn, enters the phreatic aquifer by underground lateral transfer, and the one that comes out for this last one through the sulphurous spring of the Dehesa de Ágreda.

Pyrite oxidation increases the deterioration of groundwater quality, for example, through the release of metals and metalloids such as arsenic. This is common with acidic mine waters [24]. In our case, the excess of sulphates derived from the oxidation of pyrite is what gave rise to water-supply problems: waters from the springs of Los Ojillos were used to supply water to the population of Ágreda until 1935, but due to their high sulphate content (more than 400 mg/L in low waters), they were replaced by other distant springs of better quality from the Moncayo Mountain. As the flow was insufficient in view of the increase in water demand, a complementary survey was carried out in 1983 in the marine Jurassic limestone aquifer, in the pyrite zone, which also had an appreciable sulphate content. Lastly, in 2003, a survey was conducted in the carbonate aquifer, but with a lower sulphate content, outside the main pyrite zone.

Concerning the area features, it stands up as the historical spring of Ágreda. Ágreda's sulphur source emerges in a small topographic depression where the water table outcrops towards the end of the aquifer in its northern part. It is Ágreda's historical sulphur source, whose existence has been documented in writing since the 17th century, although a Roman origin is very possible [25]. Because of its composition and properties, it was considered

to be of interest for the treatment of herpetic rash, stomach disorders (associated with its excess alkali), diuretic function (presence of sulphates), and tonic in anaemic states (because of its iron content). This source is recognizable by the intense smell of rotten eggs that it emits.

A detailed characterisation and investigation of the factors affecting pyrite oxidation is therefore crucial to reliably identify and predict the geochemical reactions associated with the evolution of water quality in the long term. Thus, the objectives of this work were:

- To define the conceptual hydrogeological and hydrogeochemical model of the two aquifers of Ágreda.
- To identify the origin of the high sulphate content of carbonate and hydrogen sulphide quaternary aquifers, and their relationship to lithology and content of pyrites and peaty sediments, assessing the importance of pyrite oxidation in aerobic and anaerobic hydrogeological environments.

## 2. Materials and Methods

The working methods followed in this study are as follows: A hydrogeological study, which consisted first of aquifer identification, an inventory of 50 water points (wells, springs, etc.), especially in and around the quaternary basin, the measurement of piezometric levels, and the specific gauging of springs. Groundwater velocity was estimated according to Darcy's law by using the results of pumping tests.

In order to know the geological environment, the nature of the quaternary aquifer of lake origin of Ágreda was completely investigated to 5 m deep, with 10 soil pits and the support of 9 Borro dynamic penetration tests. In the area between the springs of Los Ojillos and the sulphuric spring, a drainage ditch, 300 m long and 3 m deep, was cut, and four rotary drilling boreholes between 6 and 27 m deep were made available. Below 5 m, there were some data from the aforementioned drillings and penetrometers that were enough to determine the stratigraphy and facies near the hydrogen sulphide source. These scattered data from different geotechnical dispersal sources were collected by the Agreda City Council without a proper study and were combined in the current work.

To study the chemical species widely present in the springs of the study area, we conducted 35 chemical analyses that corresponded to 14 different water points. Most analyses corresponded to the most abundant and representative springs of the aquifers in the area.

Detailed full-scale groundwater monitoring was carried out over a period of nine years, sampling at three selected points along a stretch of about 1700 m in the flow direction. On this stretch, water flows 1200 m through the limestone aquifer and 500 m through the Quaternary aquifer. The selected points were:

- A supply well for the town of Ágreda that we considered at the beginning of the flow line before entering the pyrite mineralized zone.
- The discharge area of this aquifer into Los Ojillos del Keyles springs at the edge of the Quaternary aquifer (and the recharge point of this Quaternary aquifer).
- The exit point of this last aquifer (hydrogen sulphide spring of Ágreda).

In order to ascertain the compositional time variations of the groundwater, water samples were taken from both the sulphur spring and the Los Ojillos del Keyles springs over eight years (2000–2009) and on a weekly basis, except in the summer periods when the spring was drying up. The oxygen, transparency, nitrite and nitrate content, alkalinity, pH, conductivity, and temperature of the water were measured in situ, totalling 10,584 analyses. These analyses were carried out within the activities of the I.E.S. Margarita of Fuenmayor (secondary school) in Ágreda as part of the GLOBE Programme directed by one of the signatories of this article.

All analyses were carried out in the two springs (Los Ojillos spring and Ágreda sulphur source). For this, the following means were used:

- Turbidity tube with Secchi disk at the bottom to determine the transparency of the water. Values had a variability of 1 cm.
- Hanna kit (HI 3810) to quantity dissolved oxygen in water, 0–10 mg/L range, variability 0.1 mg/L.
- Hanna kit (HI 3811) to assess the alkalinity of waters. Range 0–300 mg/L; variability 3 mg/L.
- Hach Kit (Nitraver) to assess the nitrate and nitrite content of the waters. Range 0–40 mg/L; variability 0.01 mg/L
- Hanna pH meter and other brands in measuring the pH of water. Variability of 0.2 pH units.
- Hanna conductivity meter for measuring the conductivity of water; 40 μS variability.

The determination of organic C and total N was completed in Los Ojillos del Keyles springs, sulphohydric spring, and two locations near them in the Dehesa de Agreda in the Laboratory of Agricultural Chemistry and Instrumental Analytical Techniques of the School Higher Technician in Agronomic, Food, and Biosystems Engineering at the Polytechnic University of Madrid.

For the determination of the white deposits that expelled the sulphohydric source, small samples were collected on filter paper under the jet of the source for three hours. Six samples were analysed by scanning electron microscopy at the SIDI (interdepartmental research service) of the Autonomous University of Madrid.

The annual analyses of the water supply well of the city of Ágreda carried out during the period of 2003–2009 by [26] were available.

These analyses were conducted with chromatography, mass spectrometry, inductively coupled plasma, and other advanced techniques.

- An isotopic study of δ34S was performed on three water samples at the MAIMA Laboratory of the Barcelona University. Samples were analysed with a Carlo Erba 1108 Elemental Analyser linked to a Delta XP plus Finnigan Mat Isotope Ratio Mass Spectrometry (EA/IRMS) from Thermo Fisher.
- Identification in the field of pyrite mineralization and the pyrite content of the geological formations involved in the hydrogeology of the aquifers and sulphohydric springs of Ágreda and Débanos in the upper basin of the rivers Keyles and Añamaza, respectively. For this, the different geological formations in the field were walked on foot, and exhaustively for those that had an appreciable pyrite content. The pyrite content of these pyritic sedimentary formations was assessed in situ and visually, laterally observing variations in the density of the mineralization (lateral continuity). Two stratigraphic columns were also drawn in these formations or lithological columns from nearby boreholes that had been recognized.
- The cartography of the distribution of plant species that revealed the gypsum substrate in order to identify those interspersed layers with the Weald with a relative sulphate content.
- Use of PHREEQC version 3 (USGS, January 2020) software to determine the residence time in the aquifers, using the obtained field data as a starting point.

## 3. Site Geology

### 3.1. General Geological Characteristics of the Area

The area is part of the northern sector of the Aragonian Branch of the Iberian mountain range (Figure 1). In a broad sense, the stratigraphy comprises a Hercinic basement and a Permian–Triassic tegument, both of which are outcropped in the anticlinorium core of nearby Moncayo Mountain. There follow some cover formations consisting of carbonate materials from the marine Jurassic and the Purbeck-Weald Facies of a continental nature. The marine Jurassic reaches as far as the Kimmeridgian, which consists of reef limestones, on which the first sediments of siltstones, limestones, and sandstones of the Tera Group are found, in the Purbeck-Weald. Then, a predominance of carbonate materials follows, accompanied by sandstones and siltstones that form the Oncala Group. Both groups have a

thickness of more than 3000 m in the area, although the total thickness of the Weald Facies may exceed 9000 m.

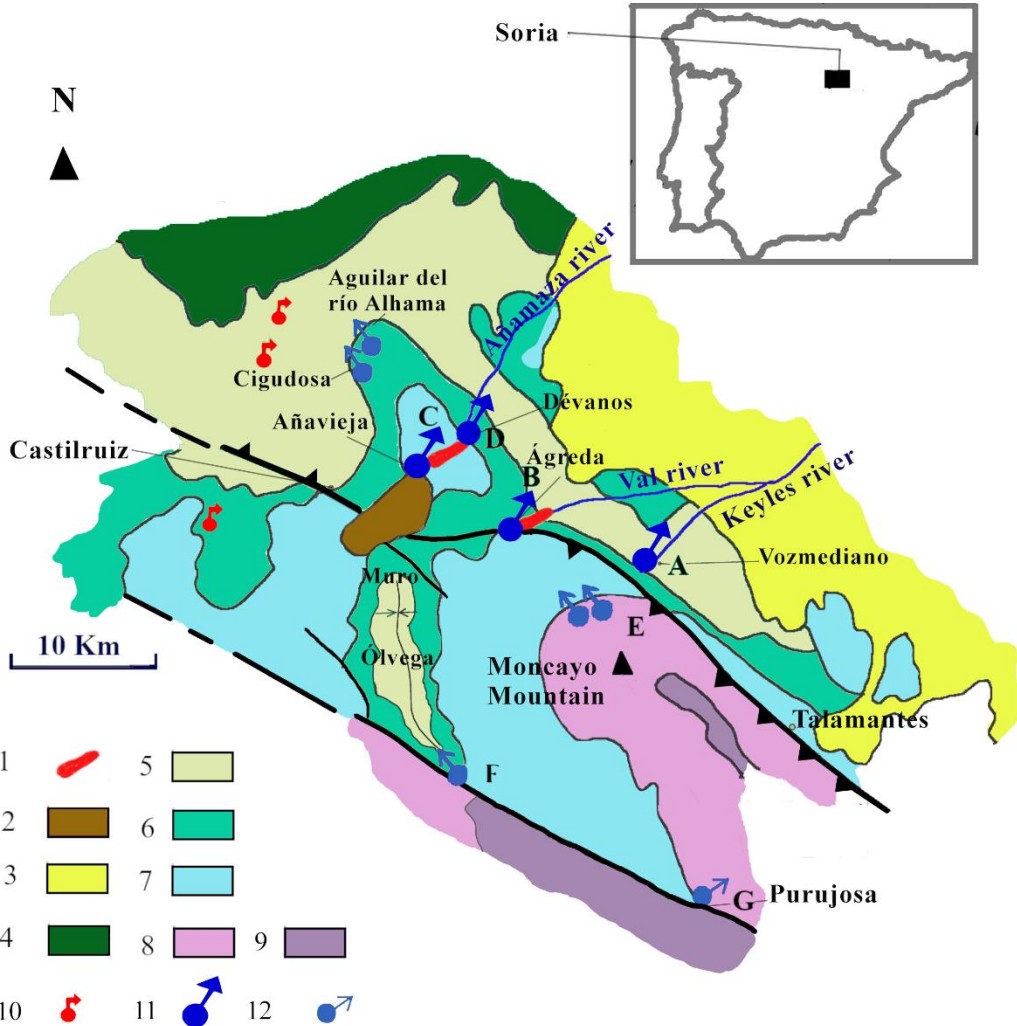

**Figure 1.** Geological setting of study area adapted from [27]. (1) Ó Calcareous tuffs and peats. (2) Añavieja Basin. (3) Tertiary molasses. Weald Facies ((4) Urbión Group. (5) Oncala Group. (6) Tera Group.) (7) Marine Jurassic. (8) Triassic (Buntsanstein facies). (9) Hercynian Basement. Springs: (10) Sulphuric springs, (11) Important springs with high relative sulphate content: A. Vozmediano (M. de Vozmediano 1, 2). B. Ojillos del Keyles,y Fuente de Ágreda. C. Añavieja (Manantial de la Lagubna II). D. Débanos (Manadero de Dévanos), H. Castilruiz (Borehole 2, 3). (12) Springs with a low sulphate content: E. Moncayo springs (Arroyo del Moncayo, 1, 2, 3. F. Vomitrosa spring, G. Purujosa spring. H. Pozo 1 de Ágreda. (13) Towns and villages.

### 3.2. Stratigraphy of Purbeck-Weald Facies in the Ágreda Area

The Tera Group in the surroundings of Ágreda begins with conglomerates with gravel-sized particles, which are nonconformant with the underlying Jurassic deposits. It follows an alternation of conglomerates, quartz sandstones, and siltstones of more than 200 m. Lastly, there is a carbonate series about 80 m thick that serves as a passage for the sediments of the Oncala Group. All these layers are part of Sequences I and II of [28]. The first corresponds to the Ágreda Fm., while the second constitutes the Magaña and Matute Fms. [29]; the first is of detrital origin, and the second of carbonate. The age assigned to these layers on the basis of stratigraphic and paleontological criteria (ostracods and charophytes) is Tithonian–lower Berriasian.

Above the Tera Group is the Oncala Group, which comprises some 2000 m of classic and carbonatic materials. This group, in the eastern part of the Cameros Basin, was divided into several formations or alloformations that form part of sedimentary sequence III of the

filling of the basin [28]. Given the lithological characteristics of the sediments [30,31] in this sub-basin, we could differentiate three major formations: Huérteles Fm., Aguilar del Río Alhama Fm., and Valdeprado Fm. The first is characterized by the predominance of clastic sediments, while the other two are characterized by the preponderance of carbonatic sediments. The Huérteles Fm. prevails in the western part of the sub-basin, while the other two do so in the eastern part (the second in the lower levels and the third in the upper part). The age of the materials, on the basis of their content in ostracods and charophytes, and by stratigraphic sequencing, is considered to be middle Berriasian.

## 4. Results: Geological and Hydrogeological Characteristics of Aquifers

### 4.1. Pyrite Distribution

Regarding the pyrite content of the area, and according to [32,33], the pyrite crystals of the Weald Facies formed as a direct consequence of the regional low-grade hydrothermal metamorphism nature that affected this eastern area of the Cameros Basin. Although all depositional sequences present small pyrite mineralisations, these appear mainly associated with the lutite levels in contact with sandstones, but not in the latter. According to [32], the sedimentary facies exert total control over the pyrite deposit: the source of Fe needed for its formation comes from the chlorites present in the lutites, and with respect to the S, it is the permeable levels of sandstones that favoured the transport and mobilisation of S-rich fluids that were in pre-existing sulphates and sedimentary pyrites. Transport and mobilisation distances of up to 1 km were estimated, but other routes of introduction of S from outside the system (Triassic gypsums, for example) through local fluid targeting sources such as shear zones and high permeability faults can also be considered.

In the area of Ágreda, pyrites are extremely abundant in the Tera group of the Weald Facies, which is formed by conglomerates, sandstones, siltstones, and limestones, where there are layers of those that had set. They are usually presented in the form of cubes with sides up to 10 cm, and in a secondary way and in other different layers in the form of dodecahedra. In the Tera Group of the Weald Facies of the Ólvega syncline, in the Keyles river drainage basin upstream from Ágreda, there are small limonitised pyritohedra. There are also pyrites in the Oncala Group above.

However, pyrite mineralisations do not only appear in the Weald facies, but they are also found in the Jurassic marine carbonate rocks of the sector located north of the parallel of Fuentes de Ágreda and which are affected by slate schistosity (Palacios, 1882; Sanz, 1981). These rocks form the Los Ojillos del Keyles aquifer. In the Jurassic rocks, they appear dispersed, concentrated in specific pyrethriferous layers, and, within these, they are sometimes concentrated in irregularly shaped pockets no more than 2 m in size. Here, pyrites are rather concentrated in the more permeable limestone layers, where they acquire density similar to that of the Weald shales (Figure 2), with crystals and subrounded nodules of iron hydroxides of up to 2 cm, product of the pyrite alteration; on other occasions, only the moulds of the nodules remain. The pyrites were also present in the areas of superficial alteration of the marliest layers, where they disintegrate with some ease. In less proportion they are found in the Plio-Quaternary Raña deposits, and in outer edge facies of the Pleistocene calcareous tufas of the river Keyles in Ágreda. They also appear in the coarse sand and gravel sediments within the sediments of the ancient lagoon of Ágreda, dragged by the river Keyles and other tributaries of the lagoon by the erosion of the carbonate rocks of the marine Jurassic present in its watershed. Nowadays, they are relatively abundant in the gravel deposits of the sediment of these watercourses, as in the river Alhama between Cigudosa and Aguilar del Río Alhama, where they are mixed with limestone and sandstone pebbles.

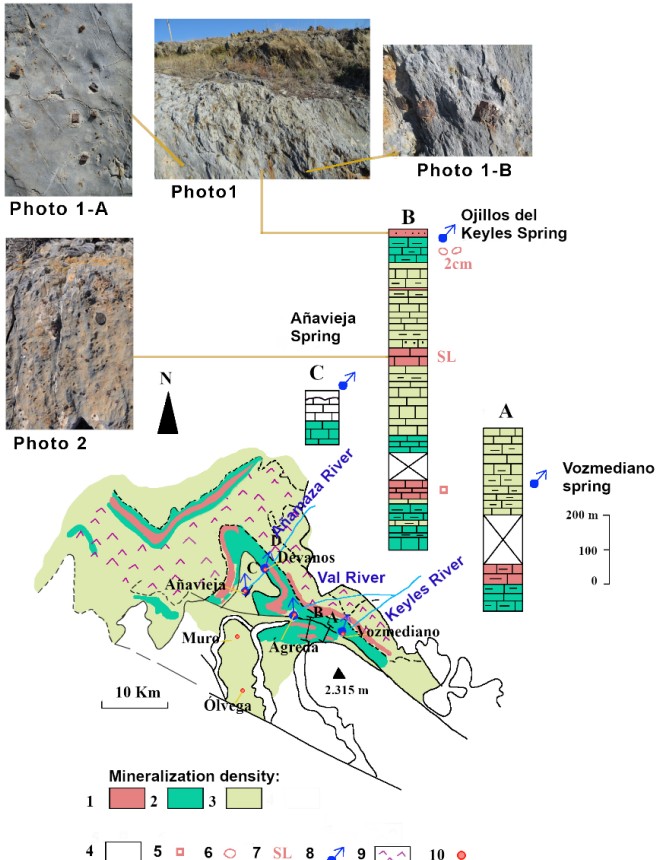

**Figure 2.** Pyrite distribution in study area. (**A**) and (**B**), Stratigraphic columns in marine Jurassic and density of pyrite mineralization. (**C**). Lithological column of the borehole next to the Añavieja spring. Mineralization density: (1) high, (2) medium, (3) low. (4) Geological formations without pyrites, (5) cubic crystals of limonitised pyrite on surface (in cm, maximal length of the sides), (6) Fe nodules and pyrite casts, (7) pyrites without limonitisation at the surface, (8) important springs with high relative sulphate content, (9) distribution of gypsiferous plant species *Gypsophila hispanic (Willk.)*, *Herniaria (8) ruticose (L.)* and *Launaea pumila (Cav.)* These plant species indicate that gypsum is clearly associated with the Oncala Group of the Weald Facies. (10) Towns and villages.

Figure 2 shows the density of mineralization in the study zone in a broad sense, including the Weald. In the marine Jurassic limestones and marly limestone within Los Ojillos del Keyles and Añavieja aquifers, stratigraphic and probing columns were added where the layers with the most abundance of pyrites and Fe nodules were identified. Although the degrees of mineralisation density in Figure 2 are arbitrary, Photos 1 and 2 can serve as a reference to show high density. The pyrites in these rocks are present as long as they are being metamorphosed, which is increasing towards the north, in the proximity of the Talamantes–Castilruiz fault zone and contact with the Weald. It is as if this tectonic accident and its zone of influence had conditioned the focalisation of fluids and favoured the introduction of external S-rich fluids from the Triassic sulphates and the dolomite, carbonate breccias, and limestones of Cortes de Tajuña, with which they are interconnected. Here, the pyrites are rather concentrated in the more permeable limestone layers, where they acquired a density similar to that of the Weald shales, with crystals and subrounded Fe nodules of up to 2 cm. There are small, nonlimonitic tiny pyrites in layers of black limestone rich in organic matter. However, pyrites are practically nonexistent in the very permeable Kimmeridgian reef limestones, represented by the symbol $J^{32}$ on the Ólvega 1/50,000th geological map [34]. This is the reason why they are sought after for the manufacture of concrete aggregates, since one of the problems of this area is having materials that are free of sulphides that could compromise the manufacture of cement and

concrete dams, and consequently the construction of public works. In the recently built concrete Enciso Dam, for example, it was key to prospect for limestone quarries without pyrite, having to go to distant areas. The same happened with the planned Cigudosa Dam, which is located in the area. In Añavieja, pyrites are also quite abundant in the compact limestones of outcrops J [24–32], oolitic, sandy, and loamy limestones [34], decreasing their abundance and size in sections $J_c$ [23,24] and J [22,23], where they barely reach the size of 1 cm; however, section J [24–32] in the area of Ólvega and Muro no longer contains pyrites.

### 4.2. Gypsum Distribution

In the Oncala Group of this area, there is gypsum, especially in the Aguilar del Río Alhama Fm., between Cigudosa and Cervera del Río Alhama. Here, it appears in layers of white gypsum no more than 20 cm thick, interbedded with tabular limestones. At some point, the waters remobilised the gypsum and precipitated it as filler for fractures. The presence of gypsum in the Weald is limited in principle to the Oncala Group, with the Tera Group being richer in pyrites.

However, it is difficult to visually identify these thin levels of gypsum, as they have largely disappeared from the surface, washed away by dissolution, although they are probably close by because of the whitish coloration in the ground. In areas where soils have a minimal gypsum content, plant species especially adapted to these types of soils grow.

Thus, shrub species *Gypsophila hispanica (Willk.)*, *Herniaria fruticosa (L.)* and *Launaea pumila (Cav.*) were recognised in the valley of the river Alhama [35] in Cigudosa-San Felices, Valdelprado, Fuentes of Magaña, and Aguilar of the river Alhama, and in the lower part of the Keyles ravine in Ágreda. The distribution of this vegetation fits to the outcrops of the Oncala formation (Figure 2).

### 4.3. Hydrogeology

#### 4.3.1. Añavieja and Débanos Springs Aquifer

Figure 1 shows the pyrite-rich marine Jurassic carbonate aquifer that drains through the Añavieja and Debanos springs (with a combined average flow of about 500 L/s [36]. These springs are found along the course of the river Añamaza, whose alluvial, as in Ágreda on the river Keyles, is formed by lagoon filling materials of an organic nature (peat and mud) over near 7 km length. There are not sufficient hydrogeological data for the whole aquifer system, but the high content of sulphate in the waters of the Devanos spring (376 mg/L) and Añavieja spring (255 mg/L [26]) and the hydrosulphuric nature of the groundwater of the quaternary aquifers observed in some probes was noted (near Devanos springs and in the alluvial aquifer, we produced a drilling borehole where 4 mg/L of $H_2S$ was obtained).

#### 4.3.2. Los Ojillos del Keyles' Carbonate Aquifer

This aquifer constitutes the NW margin of the Vozmediano spring hydrogeological system, to which there is some hydraulic connection. It is formed by marine Jurassic carbonates rocks corresponding to the Bathonian–Kimmeridgian. They are limestones, loamy and sandy limestones, limestones and sandstones and parareciferous limestones at the top of the layer, which total more than 350 m likely permeable aquifer materials (Figure 3). There are layers with abundant pyrite crystals. On this series and in contact by means of a fault to the north are conglomerates, sandstones, and siltstones of the Tera Group of the Weald Facies, there is also pyrite, which acts as an impermeable roof for the aquifer, and which is in contact with springs of Los Ojillos to the north, with an average flow of about 45 L/s, and those of Vomitrosa to the south, with an average flow of about 10 L/s (Figure 1) [37]. Los Ojillos springs are several pools of water close to each other that spring up on the northern edge of the quaternary aquifer. The water currents that originate from them rapidly gather in a channel lined with cement.

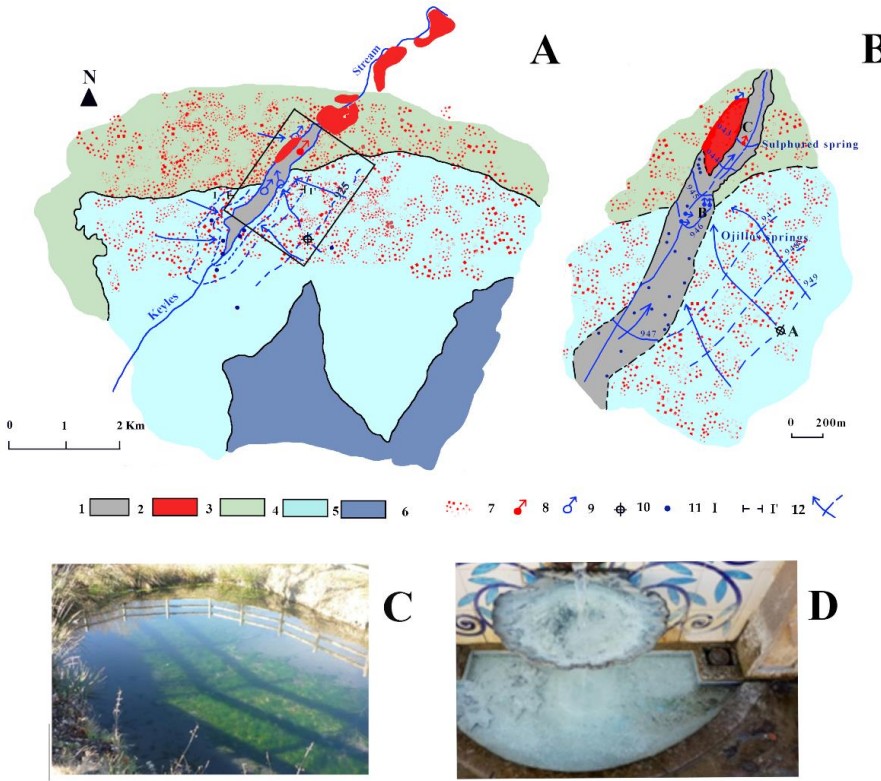

**Figure 3.** (**A**) Aquifer of Los Ojillos del Keyles springs, Ágreda hydrogen sulphide spring, and deposits of filamentous sulphates. (**A**) Location of Los Ojillos del Keyles springs and the Ágreda sulphur source. (**B**) Detail of study area. (**C**) Los Ojillos del Keyles spring. (**D**) Ágreda sulphur source and filamentous sulphur deposits. (1) Quaternary aquifer with organic sediments. (2) Calcareous tuffs. (3) Tera group of Weald facies. (4) Limestones. (5) Marly limestones and marls. (6) Pyrite mineralisations. (7) Hydrosulphide spring of Agreda. (8) Los Ojillos del Keyles springs. (9) Agreda supply well. (10) Water points. (11) Hydrogeological cut. (12) Equipotential and streamlines.

The recharge area is about 8 km$^2$ [36], occupying part of the so-called Sierra de Fuentes. Most of the recharge area consists of bare limestone hills or is covered with small holm oaks; a portion comprises dry farming fields and some orchards associated with the small fertile plain of the river Keyles; this is where the moderate nitrate content of its waters comes from (15 mg/L in the Ágreda supply survey). Average annual rainfall in the area of recharge of this aquifer is 516 mm. Natural recharge presents its maximum between December and January, and has zero values in July and August, following data time series from 2010–2021. It is a little-exploited aquifer, with flow directed towards the springs of Los Ojillos and towards the quaternary basin, where it diffuses among its sediments (Figure 3A,B). The waters of Los Ojillos have sulphate–calcium facies, which are very oversaturated waters, and for this reason they have deposited tufa throughout the quaternary there, forming water impoundment by tufa dams, which today are full of sediment. The Ágreda supply well is located in this aquifer at a distance of 1200 m to the southeast of Los Ojillos springs, enough for the groundwater to not have reached the most pyritic area, and for the sulphate content not to exceed the limits of potability (Table 1). Depending on hydraulic gradient and hydrogeological parameters obtained from the pumping test of this well, an actual groundwater velocity of 21 m/day can be estimated. This velocity was obtained by applying Darcy's law with the results of transmissivity (1500 m$^2$/day) and storage coefficient ($2.5 \times 10^{-3}$) of a pumping test carried out in the Agreda supply well, where the free aquifer has a mean saturated 70 m thickness. The natural mean hydraulic gradient of the area is considered as 0.0025.

**Table 1.** Chemical analyses of most representative springs in the study area. Summary of major hydrochemical parameters for groundwaters (Ágreda–Soria).

| Name | Date | pH | C (µS/cm) 25 °C/20 °C | Chemical-Analysis Concentrations (mg/L) | | | | | | | | | | | | |
| --- | --- | --- | --- | --- | --- | --- | --- | --- | --- | --- | --- | --- | --- | --- | --- | --- |
| | | | | $Ca^{2+}$ | $Mg^{++}$ | $NH_4^+$ | $HCO_3^-$ | $Cl^-$ | $SO_4^{2-}$ | $NO_3^-$ | $NO_2^-$ | $SiO_2$ | $HS^-$ | $O_2$ | $CO_2$ Free | TDS |
| M. de Vozmediano 1 | 29 September 1981 | 7.5 | 558.56 | 104 | 37 | | 220 | 14 | 188.4 | 15.3 | 0 | | | | | 587.7 |
| M. de Vozmediano 2 | 21 April 1984 | 6.5 | 373 | 76.2 | 29 | 0 | 188 | 14 | 143.5 | 10.7 | 0 | | | | | 465.6 |
| Purujosa | 21 April 1984 | 6.9 | 213 | 48.1 | 12 | 0 | 131 | 9.2 | 47.6 | 9.3 | 0 | | | | | 260.3 |
| Manadero de Débanos | 21 April 1984 | 6.9 | 760 | 136 | 54 | 0 | 317 | 14 | 376.3 | 6.4 | 0 | | | | | 908.5 |
| Vomitrosa | 20 September 1981 | 7.4 | 476.86 | 92.2 | 15 | | 293 | 3.5 | 47.6 | 10.7 | 0 | | | | | 451.57 |
| 352-1-4 | 24 September 1981 | 7.2 | 86.96 | 28.1 | 2.4 | | 95.2 | 3.5 | 0 | 0 | 0 | | | | | 129.59 |
| 331-4-3 (1) | 26 September 1981 | 6.9 | 40.8 | 4 | 2.4 | | 12.2 | 3.5 | 2.7 | 5 | 0 | | | | | 30.35 |
| 331-4-3 (2) | 21 April 1984 | 5.1 | 19 | 8 | 3.6 | 0 | 39.1 | 9 | 2.1 | 0 | 0 | | | | | 63.45 |
| Castilruiz Borehole 2 | 5 May 2008 | 7.4 | * 794 | 147 | 19 | | 272 | 8.7 | 216.8 | 7.48 | | | | | | 690.36 |
| Castilruiz Borehole 1 | 30 July 2008 | 7.2 | 940 | 154 | 35 | | 337 | 7.2 | 277.4 | 0.1 | | | | | | |
| M. de la laguna II | 7 April 2016 | 7.3 | 830 | 156 | 18 | <0.050 | 215 | 5.1 | 233 | 8.54 | | 10 | | 8.8 | 8 | |
| Fuente de Ágreda (ˆ) | 24 September 2015 | 7.3 | * 911 | 184 | 29.6 | 0.68 | | 14.1 | 286 | 5.2 | 0.024 | 12.4 | | 8.6 | 21 | |
| Fuente de Ágreda (ˆ) | 10 April 2013 | 7.2 | 1063 | 199 | 29.5 | 0.83 | 294 | 14.6 | 305 | 5.97 | | 11.3 | | 7.1 | 35 | |
| Fuente de Ágreda (ˆ) | 2 November 2011 | 7.1 | 1094 | 212 | 33.2 | 0.17 | 288.64 | 15.8 | 302 | 8.61 | 0.071 | 13.1 | | 7.6 | 7 | |
| Fuente de Ágreda (ˆ) | 30 September 2009 | 7.2 | 1125 | 216 | 37.4 | <0.02 | 308 | 23.4 | 425 | 9.1 | 0.56 | 12.2 | | 3.7 | | |
| Fuente de Ágreda (ˆ) | 30 March 2006 | 6.8 | | 195 | 28 | 0.79 | 263 | 18 | 271 | 4 | 0.017 | 12 | | 3.5 | 34 | |
| Fuente de Ágreda (ˆ) | 3 April 2003 | 7.3 | 870 | 180 | 32 | | 285 | 16 | 290 | 3 | | 7.2 | | 2.4 | 21 | |
| Fuente de Ágreda (+) | 12 March 1883 | | | 117 | 46 | | 347 | 49 | 160 | | | 28 | 30 | | | |
| | Min. | 6.8 | 870 | 117 | 28 | <0.02 | 263 | 14.1 | 160 | 3 | 0.017 | 7.2 | 30 | 2.4 | 7 | |
| | Max. | 7.3 | 1125 | 216 | 46 | 0.83 | 347 | 49 | 425 | 9.1 | 0.56 | 28 | 30 | 8.6 | 35 | |
| | Media | 7.15 | 969.67 | 186.14 | 33.67 | 0.62 | 297.61 | 19.29 | 291.13 | 5.98 | 0.168 | 13.74 | 30 | 5.48 | 23.6 | |
| Los Ojillos del Keyles (ˆ) | 26 November 2018 | 7.2 | 718 | 150 | 21.2 | | | | | 22.8 | | | | 5.1 | | |
| Los Ojillos del Keyles (ˆ) | 24 September 2015 | 7.2 | * 724 | 145 | 21.7 | <0.13 | | 11.5 | 221 | 26.0 | <0.02 | 10.6 | | 6.7 | 11 | |
| Los Ojillos del Keyles (ˆ) | 28 January 2013 | 7.2 | 784 | 151 | 23.2 | <0.050 | 207 | 7.61 | 229 | 15.9 | | 11.1 | | 8.0 | 25 | |
| Los Ojillos del Keyles (ˆ) | 2 November 2011 | | 875 | 140 | 22.1 | <0.050 | 206.64 | 7.57 | 211 | 16.6 | <0.001 | 9.97 | | 7.9 | 10 | |
| Los Ojillos del Keyles (ˆ) | 30 September 2009 | 7.2 | 736 | 143 | 22 | <0.02 | 209 | 10.6 | 238 | 21.7 | 0.01 | 8.7 | | 5.4 | | |

**Table 1.** *Cont.*

| Name | Date | pH | C (µS/cm) 25 °C/20 °C | Chemical-Analysis Concentrations (mg/L) | | | | | | | | | | | | |
|---|---|---|---|---|---|---|---|---|---|---|---|---|---|---|---|---|
| | | | | Ca²⁺ | Mg⁺⁺ | NH₄⁺ | HCO₃⁻ | Cl⁻ | SO₄²⁻ | NO₃⁻ | NO₂⁻ | SiO₂ | HS⁻ | O₂ | CO₂ Free | TDS |
| Los Ojillos del Keyles (^) | 30 March 2006 | 7.0 | | 156 | 20 | <0.046 | 212 | 9 | 186 | 17 | <0.01 | 11 | | 7.2 | 26 | |
| Los Ojillos del Keyles (^) | 3 April 2003 | 7.4 | 715 | 140 | 20 | | 250 | 12 | 230 | 22 | | 6.8 | | 5.8 | 7 | |
| Los Ojillos del Keyles (**) | 10 May 1989 | 7.6 | 640 | 133 | 29 | | 273 | 18 | 227 | 21 | | 9.8 | | | | |
| Los Ojillos del Keyles (**) | 23 March 1983 | 7.2 | 797 | 132 | 24 | | 262 | 7.1 | 206 | | | | | | | |
| Los Ojillos del Keyles (**) | 2 October 1981 | 7.8 | 721.1 | 132 | 46 | | 293 | 14 | 285.2 | 12.2 | 0 | | | | | 792.1 |
| | Min. | 7.0 | 640 | 132 | 26 | <0.02 | 207 | 7.57 | 186 | 15.9 | 0 | 6.8 | | 5.1 | 7 | 792.1 |
| | Max. | 7.8 | 875 | 156 | 46 | <0.05 | 293 | 14 | 285.2 | 22.8 | <0.02 | 11.1 | | 8.0 | 26 | 792.1 |
| | Media | 7.31 | 745.57 | 142.2 | 24.92 | | 239.08 | 10.82 | 225.91 | 19.47 | 0.002 | 9.71 | | 6.59 | 15.8 | 792.1 |
| Pozo 1 Ágreda (^) | 28 March 2017 | 7.5 | * 670 | 109 | 14.7 | <0.050 | 171 | 7.29 | 131 | 15.6 | | 8.0 | | 10.1 | 4 | |
| Pozo 1 Ágreda (^) | 7 April 2016 | 7.6 | 450 | 78.1 | 7.71 | <0.050 | 141 | 6.21 | 63.9 | 14.5 | <0.010 | 7.1 | | 11.6 | 8 | |
| Pozo 1 Ágreda (^) | 10 April 2013 | 7.4 | 367 | 69.9 | 6.7 | <0.050 | 128 | 8.3 | 45.2 | 20.7 | <0.001 | 6.2 | | 9.7 | 9 | |
| Pozo 1 Ágreda (^) | 27 November 2012 | 7.3 | 632 | 130 | 17.3 | <0.050 | 201 | 7.2 | 180 | 12.4 | <0.001 | 8.58 | | 8.7 | 12 | |
| Pozo 1 Ágreda (^) | 2 November 2011 | 7.2 | 697 | 139 | 19.2 | <0.050 | 205 | 7.71 | 174 | 13.2 | <0.001 | 7.91 | | 8.1 | 6 | |
| Pozo 1 Ágreda (^) | 15 April 2009 | 7.4 | 402 | 90.4 | 10.2 | <0.13 | 160 | 9.1 | 90.4 | 17.2 | <0.020 | | | 9.4 | 7 | |
| Pozo 1 Ágreda (^) | 18 December 2007 | 6.9 | | 99 | 19 | 0.077 | 204 | 11 | 101 | 21 | <0.01 | 9.3 | | 10.6 | 13 | |
| | Min. | 6.9 | 367 | 69.9 | 6.7 | <0.05 | 141 | 6.21 | 45.2 | 12.4 | <0.001 | 6.2 | | 8.1 | 4 | |
| | Max. | 7.6 | 697 | 139 | 19.2 | <0.13 | 205 | 11 | 180 | 21 | <0.01 | 9.3 | | 11.6 | 13 | |
| | Media | 7.33 | 524.0 | 102.2 | 13.54 | | 172.86 | 8.12 | 118.67 | 16.37 | | 7.85 | | 9.74 | 8.43 | |

(*) Conductivity 20 °C; (**) Yélamos y Sanz, 1994; (^) CHE; (+) Núñez y Sunier (1885).

### 4.3.3. Quaternary Aquifer of Dehesa de Ágreda

It is an alluvial unconfined aquifer of lagoon origin, and with a significant thickness along the river Keyles valley over a length of about 3 km and 200 m in average width (Figure 4). It was dedicated to livestock grazing and orchards of 80 hectares in ancient and medieval times, and as a municipal park since the 18th century. This alluvial is the filling of an old tufa barrier closing lagoon that was functional during the Middle–Upper Pleistocene and part of the Holocene but was filled by alluvium until it was completely filled in during the historical period [25]. It is crossed by the river Keyles with an average flow of no more than 100 L/s [36]. This river is a gaining stream as it passes through this aquifer since it receives the diffuse and occasional discharges of the underlying carbonate aquifer represented in this last case in the springs of Los Ojillos that are born here. The river is largely channelled through the park and the town of Ágreda to avoid overflows during flooding, so it currently interacts little with the aquifer.

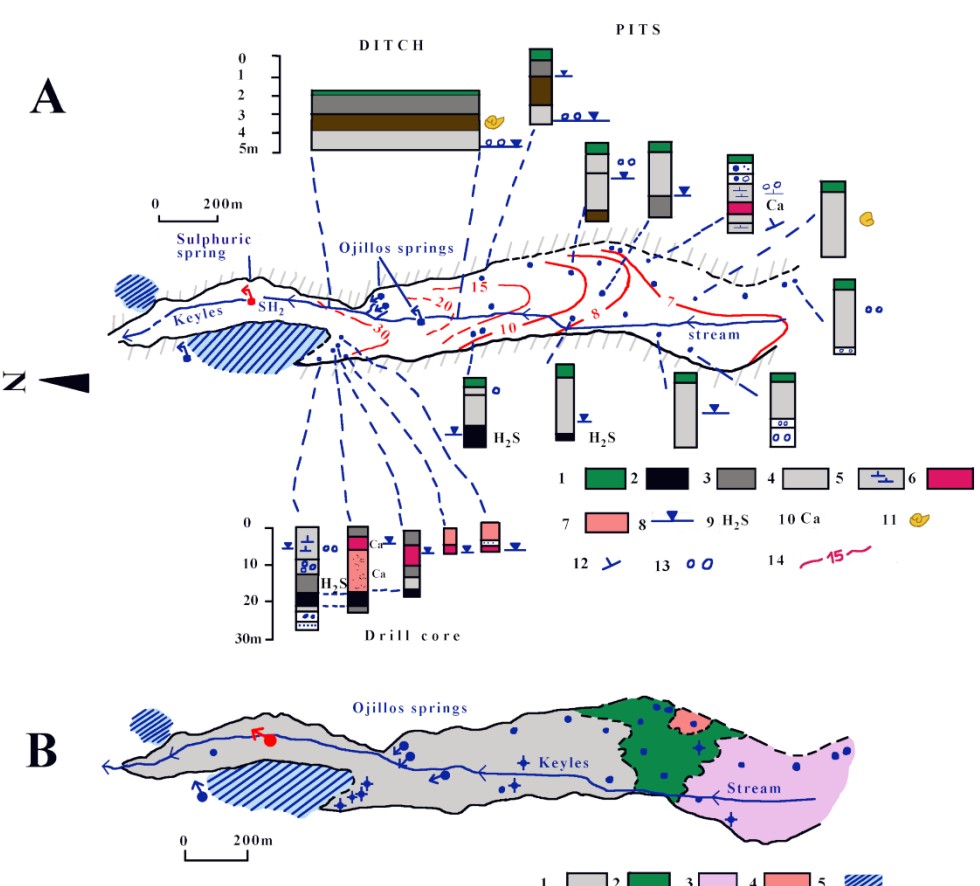

**Figure 4.** Geology of Quaternary aquifer. (**A**) (1) Soils; (2) peats; (3) clays and silts with much organic matter; (4) sandy silts; (5) calcareous silts; (6) calcareous tufa and tufa accumulations; (7) sands; (8) phreatic level; (9) sulphuric groundwater—$H_2S$ (10) calcareous sands or silts. Tufa levels, (11) gastropods; (12) plant remains; (13) levels of carbonate nodules or isolated nodules; (14) substrate depth (m). (**B**) (1) Peat and clays with much organic matter; (2) clayey silts; (3) sandy silts; (4) gravel; (5) calcareous tufas.

The recharge of this aquifer is through the lateral transfer of the calcareous aquifer and the water that diffuses from the springs of Los Ojillos. However, the quaternary aquifer is not very permeable and dynamic, and the only visible discharge from the aquifer is the aforementioned Ágreda spring (sulphide source; Figure 4B), although it is assumed that there is some diffuse outlet to the river Keyles. This spring is located towards the northern end of the Quaternary aquifer downstream and represents the end of the hydrogeochemical evolution of the same. It has an average flow of approximately 1 L/s, although in the

sampling period carried out here, it dried up in summer. This drying out in low water during these years was not so much due to a significant drop in the water table in the quaternary aquifer, less than 30 cm maximum, but rather because the catchment area had been abandoned. Subsequently, the catchment was repaired, and the spring recovered its usual original flow.

The geometry of the aquifer and the data of prospections carried out of trenches using trial pits, probes, and penetrometers reflects the filling of a valley closed by the mentioned tufa dam: an elongated basin where thickness increases from south to north, from the tail of the lagoon to the dam, where it reaches more than 30 m. This is reflected in the geological scheme of Figure 4A where, apart from the outcrops of calcareous tufa, there are the approximate isolines of the depth of the substratum; they are approximate because not all penetrometers were useful in detecting the substratum since, in several, there was rejection due to the presence of gravel. The nature of the sediments is also reflected, where we see that clays, organic silts, and peat predominate. This large amount of organic matter comes from the accumulation of aquatic plants and marshes that were at the bottom and edges of the old lagoon. In the area between Los Ojillos and the sulphur spring where there are deep probes of up to 28 m, there were peat levels up to 4 m thick and clayey silts with organic matter, but also calcareous sands, tufa accumulation, and gravel (Figure 4A).

According to the surveys carried out for the first 5 m of depth, the predominant lagoon facies consisted of clayey silts with much organic matter and many levels of peat (Figure 4B), but sandy silts and calcareous sands also appeared, including some layers of gravel.

Of the 12 hm$^3$ of loose sediments that could approximately cover this aquifer, at least 6 hm$^3$ is a mass with a large amount of organic matter readily available to bacterial action and easily oxidized, determining a critical factor for the evolution of the redox state of groundwater from oxidation conditions to reduction.

The presence of pyrites is important in this aquifer, not only in the substrate of the quaternary basin, but also in the sediments themselves. Dragging the gullies that have flowed into the lagoon provided an important load of loose pyrites as a form of placer deposit (as can be seen today in the beds of these gullies). They are also embedded in the limestone and sandstone pebbles and gravel of the riverbeds. This dispersed mass of pyrites among silts and peaty inorganic electron-acceptor oxidizing materials constitutes an important mass and is the cause of the origin of the hydrogen sulphide in the groundwater.

Because the bacterial catalysis of organic matter is typically slow, the redox state of a particular groundwater system is highly dependent on the residence time of the water in the aquifer. In this regard, the actual average linear velocity in the aquifer ($V_r$; m/day) was calculated according to Darcy's law

$$V_r = Ki/m_e,$$

where $K$ is the permeability hydraulic conductivity, $i$ the hydraulic gradient, and $m_e$ the effective porosity. $K$ data were obtained by applying Bredding abacuses to the granulometric curves of the geotechnical studies of the most involved and representative layers such as silts and peat and considering some typical values of effective porosity [38]. Hydraulic gradient 0.0025 was obtained from the isopieces map (Figure 3B) and was assumed to be equal for the two layers. According to this, the residence times in both layers necessary to cover the 500 m distance between the main discharge zone at Los Ojillos and the sulphuric spring were estimated, defined as the travel time from the recharge at Los Ojillos to the sampling point (Table 2).

**Table 2.** Estimation of transit time in organic silt and peat layers of quaternary aquifer of Ágreda. K, hydraulic conductivity; $m_e$, effective porosity; $V_r$, average linear velocity.

| Layer | K (m/Day) | $m_e$ | Vr (m/Day) | Time (Years) |
|:---:|:---:|:---:|:---:|:---:|
| Silt | 4.32 | 0.07 | 0.154 | 8.9 |
| Peat | 7.7 | 0.05 | 0.385 | 3.55 |

### 4.3.4. Hydrochemistry

Table 1 shows the chemical analyses of the most important and representative springs of the aquifer systems in the area (Figure 1). Figure 5 shows the variations in oxygen content, transparency, nitrites and nitrates, alkalinity, pH, conductivity, and temperature of the waters from Los Ojillos del Keyles and Ágreda sulphide springs, and the precipitation in Ágreda.

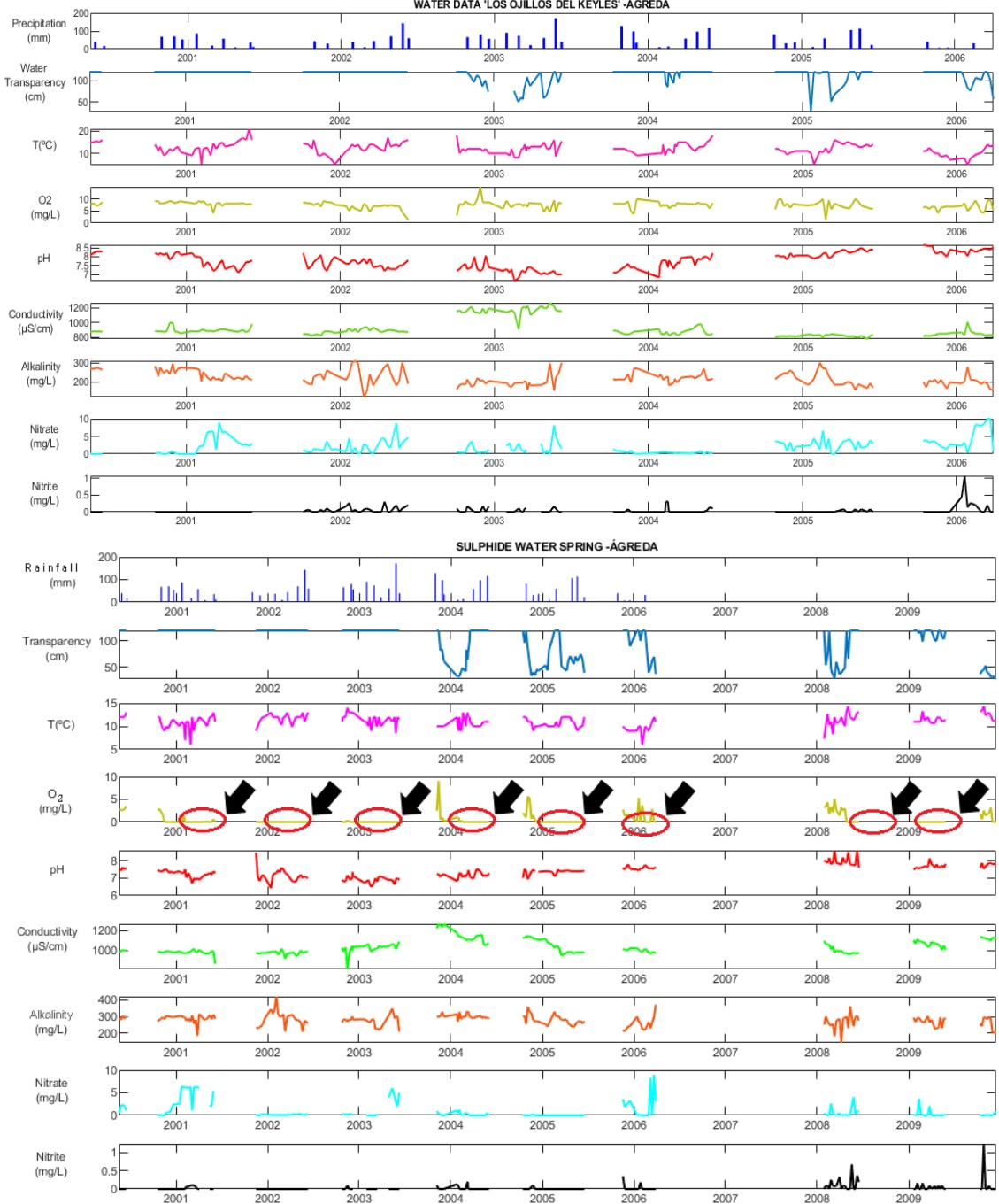

**Figure 5.** Variation of physical and chemical parameters of Los Ojillos del Keyles and Ágreda sulphide springs during 2001–2009.

## 5. Discussion

*5.1. Distribution and Alteration (Oxidation) of Pyrites*

Regarding the degree of alteration of the pyrites, the following were observed:

- Loose and fragmented pyrites in alluvial deposits are the most susceptible to weathering because they are exposed to atmospheric oxidation [39]. This is the case of the pyrites contained in the gravel beds of the alluvial quaternary aquifer of Ágreda.
- The oxidation of pyrites depends on the permeability of the rock: it is greater in limestone and sandstone than in marls, shale, and compact black limestone. In the first group, they are almost always limonitised, while in the second group, they may appear slightly altered, with bright yellow colours. In the argillaceous rocks, they are limonitised on the surface due to the effect of weathering by desquamation caused by wet–dry cycles, but slopes on roads and railways are bright at a depth of less than 2 m. Permeability depends both on the type of lithology and the presence of fractures, which are more frequent in the competent rocks of the first group. In the environment of the fractures of cuttings, pyrites look more limonitised. In the limestones, and according to drill logs in the area of Añavieja, Servicio Geológico de Obras Públicas [36], oxidation depends on the depth to which karstification and groundwater have reached; in lithological column C of a borehole next to the spring of Añavieja (Figure 2), karstification reached 41 m depth, where only iron hydroxides remain in fractures from the oxidation of the pyrites. From 41 to 95 m, there are compact limestones where some water penetration can be seen because the very small pyrites are limonitised. Below and up to at least 151 m, they are also compact limestones in which small shiny pyrites abound without oxidation.
- Smaller pyrites with crystal asymmetry or low symmetry are more easily altered [39].

The products of pyrite alteration were studied by [39], and are cited, among others, to be iron oxides and hydroxides, but the most frequent are copiapite (ferrous sulphate) and gypsum above all. This gypsum by-product of alteration and subsequent remobilisation is sometimes observed in the cracks of the Kimmeridgian limestones in their contact with the Weald facies in the area of Añavieja.

*5.2. Hydrogeochemistry: Spatial Variations and Compositional Time Variations*

5.2.1. Spatial Variations

Considering a regional scale, a high relative sulphate content was observed in the springs of the main carbonate aquifers in the pyritised zone (Añavieja springs, 376 mg/L; Los Ojillos del Keyles, near 226 mg/L; Débanos, 376.3 mg/L; and 166 mg/L data average in the spring of Vozmediano (Table 1)), and a low content in those springs that are outside the area of influence, although they drain the same geological formations (spring of Purujosa, 47.6 mg/L; Vomitrosa, 47.6 mg/L).

With regards to the Los Ojillos limestone aquifer and associated quaternary, the lack of enough sampling in the existing surveys prevents us from knowing in detail the spatial variations in the composition of the water. In the quaternary aquifer, the boreholes and trial pits had a geotechnical nature, and hardly any water samples were taken. However, hydrogen sulphide was detected in almost all surveys in the western area (more than 1 mg/L). It was also historically detected in many basement excavations in the buildings of Ágreda built on the western edge of the Dehesa, the opposite side to where the groundwater enters. However, the disposition along the same flow line defined by the Ágreda water supply borehole, Los Ojillos springs and the sulphuric spring, is enough to know the hydrogeochemical evolution of the aquifer (Line A–B–C in Figure 3B). The existence of an important historical collection of data from several years of these water points, representative of the inputs and outputs of groundwater between the two aquifers of the system, contributes to the quality of the available information.

The ionic species distribution of Ágreda hydrogen sulphide sources is represented in Table 1 and Figure 5; a large part of the values in Table 1 are presented in a binary diagram. It separates the values quite well, grouping them into the three water points (circles in

Figure 6), showing that there are changes when the water passes from one point to another. It consists of calcium bicarbonate facies, with sulphate and magnesium as the second cation and the majority anion. Regarding the appearance of Los Ojillos del Keyles, it can be observed that the chemical quality of this spring is very similar: calcium bicarbonate or calcium sulphate bicarbonate waters, with magnesium as the second major cation [40]. In order to compare the composition of both springs, and according to the data in Table 1, the composition of Los Ojillos water, and thereby the composition of water that flows to the quaternary aquifer by lateral transfer, contains dissolved oxygen of between 7.5 and 12.5 mg/L, about 15 mg/L in nitrates and between 260 and 270 mg/L in sulphates. It is clear that the oxygen is being consumed in favour of the generation of sulphates from the Agreda well 1 to the Sulphur Source (Figure 6)

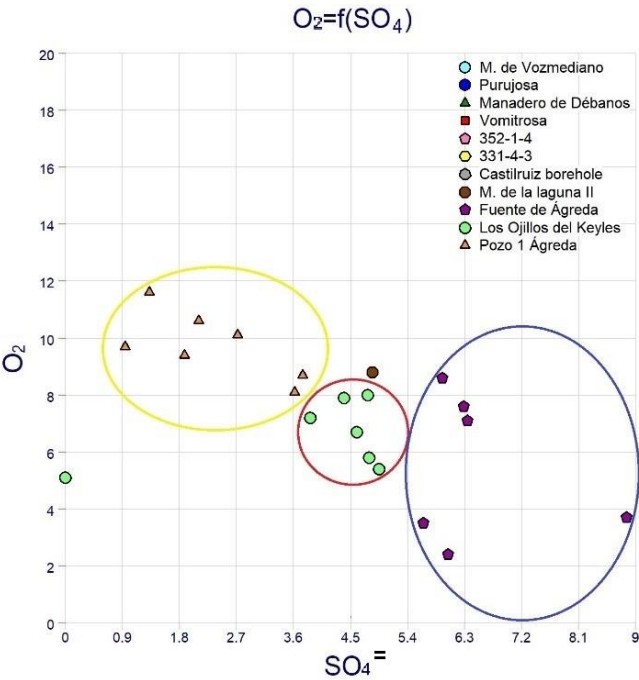

**Figure 6.** Oxygen/sulphate concentration binary diagram of a large part of the values in Table 1. Circles showing the grouping of these values according to the three-water point, and that there are changes when the water flows from one point to another. It is clear that the oxygen is being consumed in favour of the generation of sulphates from Ágreda well 1 to sulphur source.

Regarding the isotopic study, the values of $\delta^{34}S$ in the three water samples were similar, ranging from 11‰ in the Soria 1 sample to 12.5‰ in the Soria 2 sample. The result for the Soria 3 sample was 12.1‰. These analyses were conducted to determine the origin of the sulphate dissolved in the water.

### 5.2.2. Compositional Time Variations

Figure 5 shows that there is no clear relationship between rainfall variation and measured physicochemical parameters, especially in the quaternary aquifer, which confirms, to some extent, the little influence of autogenic natural recharge in this system, which is mainly fed by lateral underground transfer from the limestone aquifer.

Comparing the water from both springs, the temperature of the waters of Los Ojillos varies depending on the time of year. It also does at the Sulphhydric Fountain, but in this case, variation is very small (normally 2–4 °C, in the range of 9–13 °C). In the water of the springs of Los Ojillos, the pH is more basic (7.5–8). Its waters contain many carbonates, much more than those of Los Ojillos. The waters of Los Ojillos have a low nitrate content, around 3 mg/L, and an absence of nitrites; in the sulphite water, there is hardly any nitrate

or nitrite content, especially when there is no oxygen in the sulphurous water. The last one stands out for its great conductivity.

In Los Ojillos, there is always a large amount of oxygen, while in the sulphur source, the amount of $O_2$ is very small or zero (0, maximum 3, 5 mg/L), and always coincides with when it starts to sprout after the low water.

Turbidity was observed to be due to elemental sulphur particles. Los Ojillos water is always transparent. This is not the case with the hydrogen sulphide source, which is only transparent for a short period of time when it starts to flow after the low water. Its great initial transparency gives way to a turbidity as a result of the presence of fibrous sulphur particles. To know the nature of these particles, scanning electron microscopy was carried out which shows that practically everything is accumulations of elemental sulphur. A representative result is shown in Figure 7.

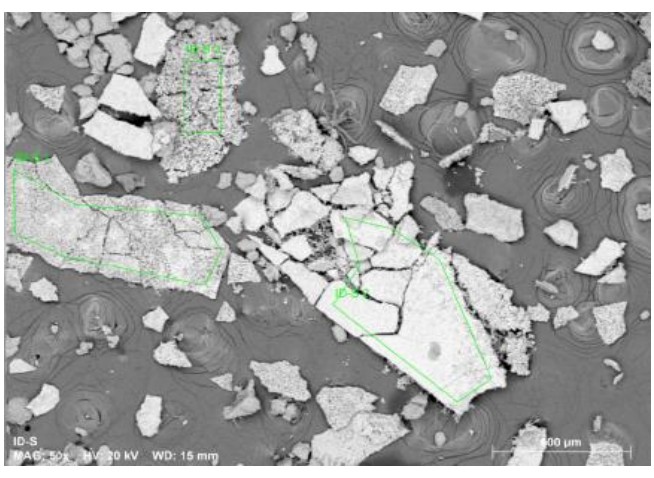

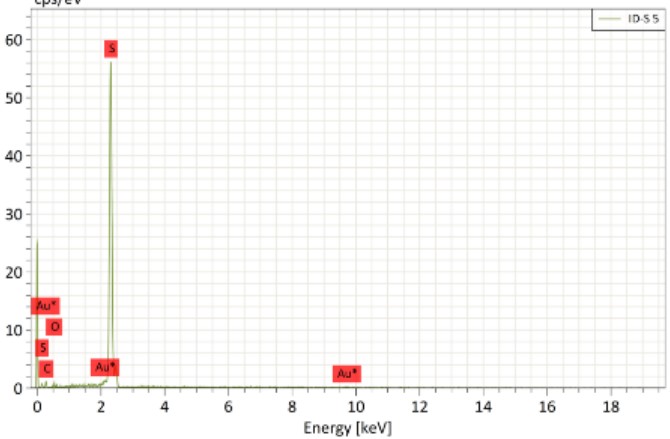

**Figure 7.** Microphotography and X-ray diffraction microscopy (EDX) of a sample taken from the sulphurous source.

Variation in alkalinity and conductivity data is not clear in the waters of Los Ojillos and in the sulphurous source, that is, with oxygen in Los Ojillos and no or almost no oxygen in the sulphurous source. It seems that with the waters with oxygen, alkalinity drops a little, while conductivity increases a little, but this does not always happen.

### 5.3. Conceptual Model of Hydrogeochemical Evolution

5.3.1. Hydrogeochemistry in the Carbonate Aquifer: Increase in Sulphates

The dominant sources of $SO_4^{2-}$ in groundwater in semiarid areas such as this are atmospheric precipitation, and the erosion of rocks and minerals [41]. According to the chemical composition of the water from streams fed directly by rain and snow from nearby Moncayo Mountain (Table 1, Moncayo streams), the $SO_4^{2-}/Cl^-$ ratio ranges from 0 to less

than 0.3. This ratio rises to 5 and to 13 for springs with a low sulphate content (Purujosa and Vomitrosa springs, respectively), and between 11 and 28 for those with a high content. The $SO_4^{2-}/Cl^-$ ratios in the latter case are much higher than the atmospheric contribution, indicating the additional contribution of sulphur from sources other than precipitation. This is a clear sign of the local origin of the sulphate ion. The concentration of $SO_4^{2-}$ in groundwater can be modified by human activities, such as the use of agricultural fertilisers. However, this activity can be considered to be quantitatively insignificant because the area of recharge is dominated by first scrubland, and then by dry farming. The possible potential sources of the sulphate dissolved in the waters of the aquifer could then be (1) pyrite oxidation or (2) the dissolution of possible gypsum present in the soil. In the previous sections, we ruled out the presence of gypsum in the recharge area of this aquifer, only present in the Oncala Group of the Weald facies. We considered it appropriate to confirm this with the signal of the $^{34}S$ isotopic composition of the groundwater.

The $\delta^{34}S$ values of sulphur dissolved in the water, which range between 11‰ and 12.5‰, fit well within the isotopic composition range of the pyrite of the Cameros Basin. Figure 8 shows the isotopic composition of the water samples and the δ34S values of the potential sources of sulphur around the area of study. $\delta^{34}S$ of Triassic gypsum ranges between 13‰ to 15.8‰ while Berriasian sulphate ranges from 15.5‰ to 20.3‰ [42]. $\delta^{34}S$ values of the pyrite in different parts of the Cameros Basin range from −8‰ to 5‰ except for the Canadillas (5–10‰) and Yanguas (11–14‰) deposits [32]. The closest pyrite deposit to the area of study was Ambasaguas, where $\delta^{34}S$ ranged from 2.6‰ to 12.4‰ [43].

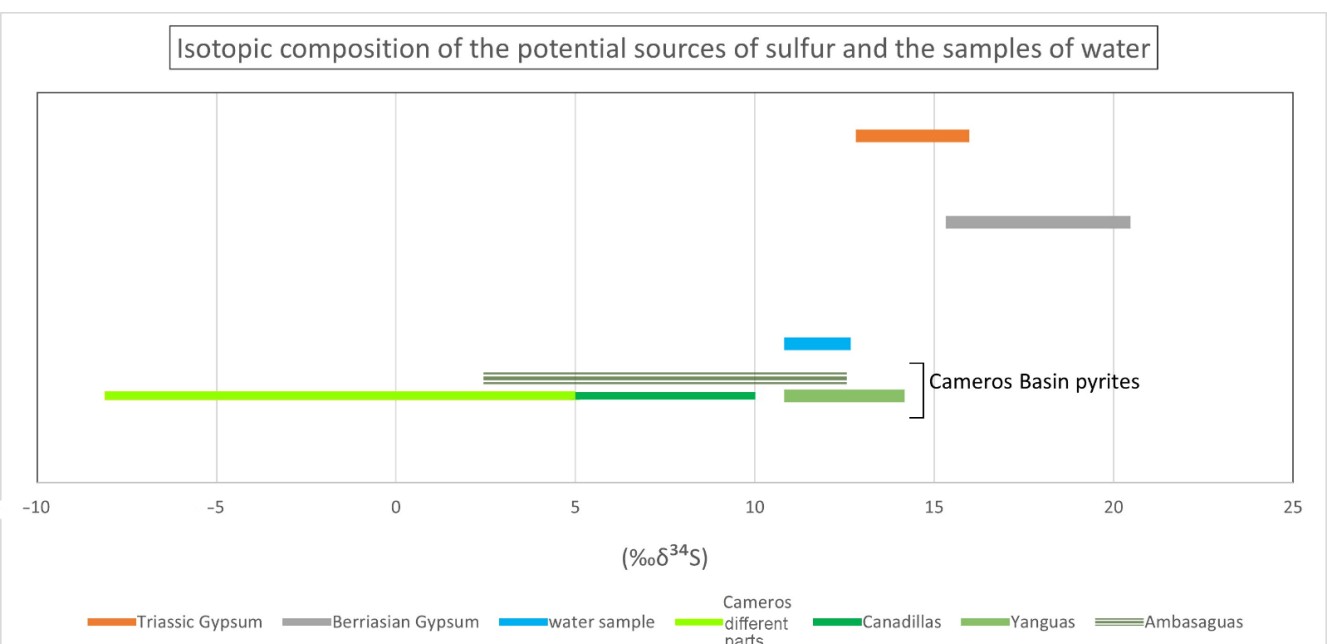

**Figure 8.** Isotopic composition of potential sources of sulphur and samples of water. Values of the $\delta^{34}S$ isotope in ‰ are graphically represented in samples of pyrites, gypsum (Berriasian and Triassic, in grey and orange, respectively), and water (in blue). Relationships in light green correspond to pyrite samples from the Cameros Basin; in lime green, the values of different zones are reflected; pyrite samples studied from the localities of Canadillas and Yanguas [32] (in alpine green and pastel green respectively) reflect higher values. Isotopic ratios in ‰ $\delta^{34}S$ obtained from the Berriasian and Triassic gypsum are higher than those of the analysed water samples. The ‰ $\delta^{34}S$ values of the analysed water samples (11–12.5) would be in the range measured in the pyrites of the Cameros Basin, specifically with those located near the town of Ambasaguas, close to the study area. For this reason, this area is considered as a possible source of sulphur for the analysed waters.

Considering a regional scale, the high relative sulphate content of the springs of the main carbonate aquifers in the area and reflected in (Figure 1) and Table 2 (Añavieja, Los Ojillos del Keyles and Vozmediano springs) is due to the existence of appreciable mineralization of dispersed pyrite. The gypsum of the Oncala group in the Weald facies does

not participate in the recharge of the mentioned aquifers either as autogenous recharge or as allogenous, and remains in the periphery of the same (Figures 1–3). The sulphate content in the waters of the aforementioned springs is proportional to the pyrite-mineralised surface of their aquifers (Figure 2). The Vozmediano aquifer can also be influenced by the introduction of recharge water from permanent and ephemeral streams in the Moncayo mountain, which are rich in oxygen, as occurs in many other aquifers [44], unlike the Los Ojillos aquifer, where recharge is only self-generated. In Los Ojillos del Keyles aquifer (206.64 to 293 mg/L), for example, if we compare the sulphate content with that of the Purujosa spring (10 L/s on average) (47.6 mg/L) which drains this same aquifer of Jurassic limestones, but outside the zone of influence of pyrites (Figure 2), we see that its sulphate content is 47.6 mg/L, i.e., some 6 times less. If we compare it with the Vomitrosa spring (35.2 mg/L), which also drains the Kimmeridgian limestones outside the pyrites' zone of influence, it is seven times less.

On a more local scale, Figure 9 shows the hydrogeological profile in the carbonate aquifer along a flow line, from the Ágreda supply well to its discharge at Los Ojillos del Keyles springs under oxic conditions; and then from this spring to the hydrogen sulphide spring, under reducing conditions. The increase in sulphates from A to B is a product of the oxidation of pyrites, and oxygen and nitrates are consequently decreased, being consumed in this reaction. From B to C, oxidation of the pyrite continues, causing a further decrease in oxygen. At the same time, there is an appreciable decrease in nitrate content. Furthermore, the existence of organic matter in the sediment passed through leads to the production of hydrogen sulphide and, occasionally, to the formation of elemental sulphur.

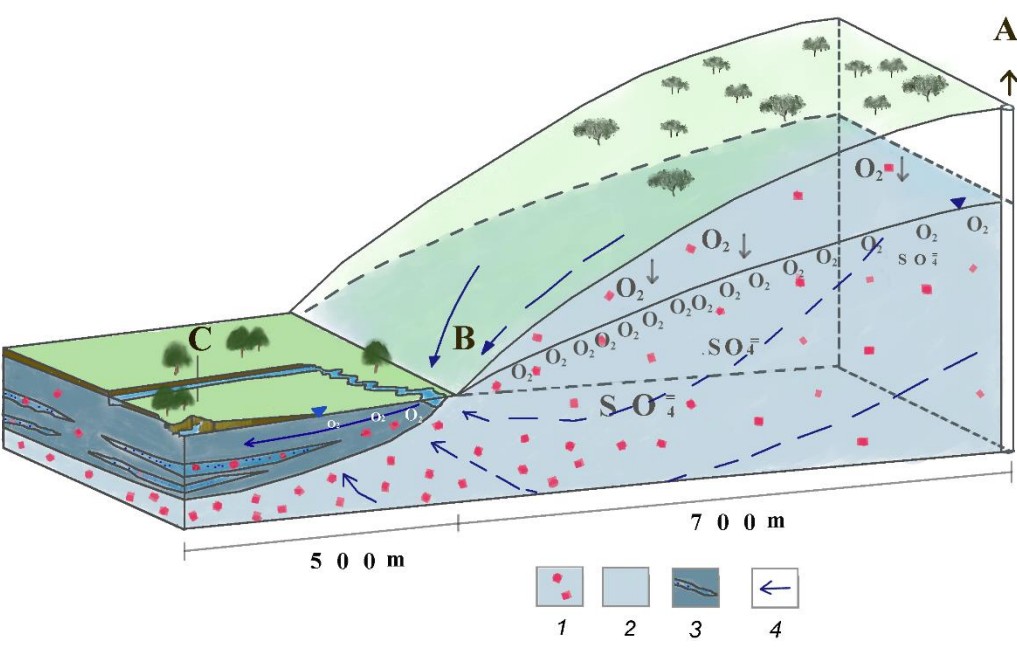

**Figure 9.** Hydrogeochemical model represented in diagrammatic sections along A–B–C flow line (Figure 3B) (A–B in the calcareous aquifer, and B–C in the quaternary aquifer with organic sediments).

Comparing the chemical analyses, at a distance of 1200 m from the well to the springs of Los Ojillos del Keyles, changes were detected in the system that translated into an increase in dissolved solids in general, but above all an increase in sulphates (from 112 to 226 mg/L average), simultaneous with a slight increase in nitrates (from 16.4 mg/L average in the well to 19.6 mg/L in the springs) and a decrease in dissolved oxygen (from 9.74 mg/L average to 6.6 mg/L) (Figure 6). This provides evidence that pyrite oxidation is the dominant source of sulphate in the aquifer, as the flow enters the most mineralized zone of the aquifer (Figures 3B and 9). Considering that the average real velocity of the groundwater in this aquifer is 21 m/day, from the Ágreda supply borehole to the Los

Ojillos spring would take 57 days, there is a reduction in dissolved oxygen of 3 mg/L and a nitrate increase rate of 3.2 mg/L/km.

For both the unsaturated and the saturated zones, with or without bacterial involvement, oxidation reactions may be due to [45]:

$$2FeS_2 + 7O_2 + 2H_2O \rightarrow 2Fe_2 + 4SO_4{}^{2-} + 4H^+ \tag{1}$$

$$2Fe_2{}^+ + 1/2O_2 + 2H_2O \rightarrow Fe_2O_3 + 4H^+ \tag{2}$$

or according to the final reaction (Reed, 1984).

$$FeS_2{}^- + 15/8O_2{}^- + 13/2Fe^{3+} + 17/4H_2O \rightarrow 15/2Fe^{2+} + 2SO_4{}^{2-} + 17/2H^+ \tag{3}$$

The oxygen present in the atmosphere and the natural recharge water in winter are the main oxidizing agents, the dissolved $CO_2$ being the accelerator of these processes. The main products resulting from these oxidations are sulphates and ferrous ions, which are the origin of future precipitations of sulphates and iron oxides that are observed in the form of residual Fe nodules in the rock and in fracture fillings.

This oxidation occurs mainly in the more permeable limestone layers where the mineralisation and limonitisation of the pyrites is observed to be higher. This oxidation of pyrite for sulphate production in the saturated zone leads to a consumption of dissolved oxygen in the aquifer waters. This availability of pyrite as an electron donor in the absence of organic matter and glauconite in the aquifer, is also the cause of the water denitrification process and consumption of dissolved oxygen that contributes, together with pyrite oxidation, to the system having moderately anaerobic conditions near the discharge point [23]. Considering that the velocity of the groundwater is 21 m/day, and the transit time is 53 days for covering 1200 m, it does not explain the sulphate content but rather the local mineralogical characteristics.

This fact was confirmed with calculations made with the PHREEQC program, which, after characterising the aquifer with the initial field data and knowing kinetic reactions speeds, the transit time was 2.2 months.

The oxidation of pyrite adds hydrogen ions to the groundwater, which in turn lowers the pH. Despite the significant contribution of $SO_4{}^{2-}$ by pyrite oxidation in this aquifer, the pH values are neutral. If this expected decrease in pH does not occur, it is because the aquifer is made of limestone, with hardly any dolomites. This should produce a buffering of the acidity, favouring the dissolution of the rock and an increase in carbonate ions in the water. This is a process that was observed in other aquifers, such as in [16].

### 5.3.2. Hydrogeochemistry in the Hydrogen Sulphide Aquifer of Ágreda
Hydrogeochemical Evolution: Pyrite Oxidation

The close relationship between the chemistry of the waters of Los Ojillos and the hydrogen sulphide source of Ágreda indicated in Section 5 indicates that the origin of the groundwater body of the quaternary aquifer comes from the underground hydraulic transfer of the calcareous aquifer, with hardly any influence from the river Keyles that crosses it as the winning river, and which has been channelled for decades. The influence of autogenous recharge of the aquifer itself is very small, since neither the surface of the aquifer nor irrigation has any entity. The chemical composition of the source water entering the quaternary aquifer is therefore like that of Los Ojillos del Keyles spring.

Thus, on the basis of this chemical composition, the circulation of the flow through the organic sediments of the quaternary aquifer causes changes to occur within the system where the present microorganisms preferentially use dissolved oxygen. Concentrations of dissolved oxygen in groundwater are generally in the range of 0–10 mg/L. Although this low concentration also indicates polluted water, in the current case, there is no contamination. The river is canalised and does not interact with the groundwater. The only source of water pollution is upstream in the city of Olvega, which may be eventual or occasional,

but the river is canalised. On the other hand, water from the calcareous aquifer is not contaminated and it is drinkable; therefore, the process is produced by natural groundwater with dissolved oxygen concentrations of 7–10 mg/L, which is said to be completely saturated and heavily oxidized (Los Ojillos del Keyles), while those with 0 mg/L dissolved oxygen are said to be completely depleted (sulphuric water). Water that enters the aquifer by lateral transfer has a high oxygen content at the beginning (between 5 and 8.0 mg/L) in the Los Ojillos zone to near 0 mg/L (although it varies throughout the year) in the hydrogen sulphide source 500 m away (Figure 9). Thus, in the beginning, upstream, the bacterial action has a supply of oxidants for the decomposition of organic matter. The rate of oxygen consumption in this case depends very much on the abundance of pyrite as an oxygen-consuming mineral and the availability of organic components and nutrients that need to consume oxygen for activity.

Decreased dissolved oxygen indicates increased biological activity. As the oxygen tends to be consumed through the aquifer, the bacteria pass to other electron acceptors such as nitrates ($NO_3^-$) and sulphates ($SO_4^{2-}$) [23]

$$2CH_2O + SO_4^{2-} + H^+ \rightarrow HS^- + 2H_2O + 2CO_2 \qquad (4)$$

In turn, the bisulphide ion passes into the characteristic hydrogen sulphide gas through the reaction:

$$HS^- + H^+ \rightarrow H_2S \qquad (5)$$

Thus, it is the organic matter in the presence of excess sulphates that triggers the production of hydrogen sulphide.

On the other hand, water experiences a decrease in alkalinity in the sulphuric source with respect to the starting water represented by Los Ojillos.

Pyrite oxidation in aerobic environments by microorganisms is well-researched; under anoxic conditions, however, the issue is open, although it seems that microbial pyrite oxidation can occur via nitrates through electron acceptors and sulphate production [7,20], as follows

$$14NO^{3-} + 5FeS_4^{2+}H^+ \rightarrow 7N_2 + 10SO_4^{2-} + 5Fe^{2+} + 2H_2O \qquad (6)$$

Pyrite is present not only in the substrate, but also in the detrital sediments of the small tank. This pyrite could be unstable in moderately reducing (anionic) conditions of certain sectors of the aquifer, and that could be the cause of an increase in sulphates from this spring with respect to Los Ojillos.

Because of all these causes, oxygen tends to be consumed as the groundwater moves through the aquifer and, by the time the water leaves the source, it is sometimes already completely depleted.

On the other hand, residence time is not too long (it varies between 3.55 and 8.9 years, and as was possible to verify with the PHREEQC program using the same process as for the carbonate aquifer, the transit time was obtained as 4.1 years) for the ecological succession of the electron acceptance terminal processes to occur either, probably because the abundance of pyrite is very great, and it is counteracted. The $O_2$ reduction rate is at least 1.4 mg/L every 100 m or, expressed in time, between 0.7 and 2 mg/L/year. If we compare it, for example, with other aquifers [2], this reduction is very fast in this surface aquifer, since oxidation is linked more to sulphur oxidation than to carbon oxidation.

Hydrogen sulphide caused by sulphate reduction passes through the zone with organic matter, and can be transformed into sulphates again (directly: reaction (7); in two stages: reactions (8) and (9), or into sulphur elemental (reaction (8)). Under slightly alkaline pH conditions, with sufficient oxygen, reaction (7) is the most favourable energetically, in the absence of thiosulphates (10). However, when oxygen concentrations are very low (as seen in the analysis of the waters of the sulphur source, where it even disappears; Figure 5), reaction (8) takes place. For this reason, when the water from this source has enough

oxygen, no sulphur precipitation is observed, with the water being very transparent; the opposite occurs when the concentration of $O_2$ greatly decreases.

$$H_2S + 2O_2 \rightarrow SO_4{}^{2-} + 2H^+ \tag{7}$$

Final product, sulphate, $\Delta G_0 = -798.2$ kJ/reaction.

$$HS^- + \frac{1}{2}O_2 + H^+ \rightarrow S^o + H_2O \tag{8}$$

Final product, sulphur, $\Delta G_0 = -209.4$ kJ/reaction.

$$S^o + H_2O + \frac{1}{2}O_2 \rightarrow SO_4{}^{2-} + H_2O \tag{9}$$

Final product, sulphate, $\Delta G_0 = -587.1$ kJ/reaction.

$$S_2O_3{}^{2-} + H_2O + 2O_2 \rightarrow 2SO_4{}^{2-} + 2H_2O \tag{10}$$

Final product, sulphate, $\Delta G_0 = -818.3$ kJ/reaction.

As the means is slightly alkaline (see pH data), reaction (8) does not take place, and no sulphur precipitate is observed. Therefore, reaction (9) would not take place either. The presence of thiosulphates is not to be expected either, so the normal reaction should be (7).

Since the pH is normally very slightly basic, the appropriate bacteria are *Proteobacteria* (especially *b-Proteobacteria*, given the pH of the water) in carbonate systems such as the Los Ojillos aquifer, and alkalinity plays a determining role, but in the quaternary aquifer, the presence of $H_2S$ determines the low pH.

Denitrification

The nitrate amounts in Los Ojillos and Ágreda sulphur source are both low, although it goes from 3 to 1 mg/L (and sometimes 0 mg/L) in the transition from one source to another. In this 500 m journey, which takes between 3.5 and almost 9 years, there is a denitrification rate of 0.3 mg/L per 100 m. This slight decrease in the nitrates of this spring with respect to Los Ojillos is mainly caused by the presence of organic sediments as the main electron donor to promote this process. As shown in Table 3, the nitrates in the sulphurous source are less than those in Los Ojillos, and it is the only place where there is organic matter. This suggests that organic matter is the cause of this denitrification.

**Table 3.** Analysis of organic matter and nitrogen compounds in the waters of the Ágreda area.

| Sample | mg/L C Organic | mg/L N |
|---|---|---|
| Dehesa de Agreda | <0.1 | 8.40 |
| Dehesa Park | <0.1 | <0.2 |
| Fuente de Ágreda (sulphurous source) | 0.74 | 3.30 |
| Los Ojillos | <0.1 | 5.68 |

This requires the presence of dissolved oxygen at concentrations higher than 2 mg/L [2], which occurs in the area of Los Ojillos on the eastern edge (7.5–12.5 mg/L). However, complete denitrification does not occur because the oxygen is consumed in pyrite oxidation, that is, oxygen was mainly used to oxidise the pyrites and produce sulphates. The smell of hydrogen sulphide is a consequence of the existence of sulphides still in the water. As the pH there is almost neutral, both sulphur and hydrogen sulphide (much more sulphur) coexist as species.

On the other hand, the reactions with a greater decrease in free energy are those that are executed as a priority. The data show that pyrite oxidation is a more favourable reaction

($\Delta G_0 = -798$ kJ) than denitrification ($\Delta G_0 = -476$ kJ). This suggests that there must have been pyrite oxidation from the beginning.

Compositional Time Variations

Normally, the first analyses of each year must wait until the source begins to flow, since, as was observed some time ago, when summer arrives, the water stops flowing and does not flow again until mid-October (Figure 5a,b). At the beginning of the resurgence, its waters contain dissolved oxygen, but it later disappears. When observing the data of several years (2002, 2003, 2008, and 2009), the periods with very clear water correspond above all with the presence of oxygen, which occurs in a short period of time when it begins to flow after the low water, while waters with little transparency show an absence of oxygen. The high initial transparency gives way to turbidity as a result of the presence of sulphur. Its waters contain many carbonates, in much higher amounts than those of Los Ojillos; however, its pH is not alkaline, like the latter. The situation is easily explained if we consider their neutralisation by the other components they possess (hydrogen sulphide, for example). It is worth noting the small temperature variation they show throughout the year (9–13 °C), the generally low content of nitrates and nitrites, and their high conductivity. The variation in alkalinity and conductivity data is not clear in waters with and without oxygen in the hydrogen sulphide source. Against oxygen presence, alkalinity decreases slightly, and conductivity also slightly increases.

## 6. Conclusions

Interdisciplinary studies in hydrogeochemistry are very useful to advance the knowledge of hydrogeological conceptual models, for example, in hydrothermal systems [46], and to know the sources of nitrates in groundwater, for example, in [47], where statistical techniques were applied to relate chromium with nitrates in agricultural areas.

Along a 1200 m stretch of fast underground flow in the zone close to the discharge into a limestone aquifer, pyrite oxidation was the determining factor that explained the increase in sulphates and decrease in oxygen and high denitrification rates. In addition, the decrease in pH, which generally accompanies pyrite oxidation, may have been masked by the high pH neutralisation capacity of the carbonate minerals. This flow then continues for about 500 m at a much slower rate through an aquifer rich in organic sediments and pyrites. Peat and pyrite oxidation in an anaerobic environment lead to the cancellation of dissolved oxygen, high denitrification, the production of hydrogen sulphide, and the generation of elemental sulphur. Thanks to intensive geological field work and prospecting, both aquifers were characterised in detail. These processes, which occur in such peculiar hydrogeological systems, were observed and measured over an extended period of time of about nine years, at three representative points along a flow line. This is an example of an exceptional natural hydrogeological environment that provides guidance on certain hydrogeochemical processes.

**Author Contributions:** Conceptualization, E.S. and C.P.; methodology, E.S.; software PHREEQC, C.B.; validation, E.S. and I.M.P.; formal analysis, E.S.; investigation, E.S.; resources, I.M.P.; data curation, C.P.; writing—original draft preparation, E.S.; writing—review and editing, I.M.P.; visualization, C.F. and I.M.P.; supervision, I.M.P.; project administration, I.M.P.; funding acquisition, E.S. All authors have read and agreed to the published version of the manuscript.

**Funding:** This research was partially funded by Spanish Ministry of Economy, Industry and Competitiveness (Ministerio de Economía, Industria y Competividad de España), grant PID2019-106887GB-C33, and financed by Madrid Polytechnic University grants VAGI18ESP and GI1504350086-2019.

**Acknowledgments:** We sincerely thank the students involved in the GLOBE Program of the I.E.S. Margarita of Fuenmayor in Ágreda, with whose help we have been able to perform the physico-chemical analysis of the springs of Los Ojillos and the sulphydric of Ágreda over the course of nine years, which is reflected in this work. We would also like to thank the Ágreda Town Council and the

Education Department of the Castilla and León Regional Government for having provided us with the reports on the geotechnical studies of the Dehesa de Ágreda park and industrial area.

**Conflicts of Interest:** The authors declare no conflict of interest.

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
