# Peer review of "Hydrogeochemical Evolution of an Aquifer Regulated by Pyrite Oxidation and Organic Sediments"

_water, doi:10.3390/w13172444_

Round 1

Reviewer 1 Report

Water

Manuscript Number: 1340798

Title: Hydrogeochemical evolution of an aquifer regulated by pyrites oxidation and organic sediments

Article Type: Research Paper

Keywords: Redox processes. Groundwater, Nitrate attenuation; Pyrite Oxidation, Peat sediments.

The authors provide a detailed full-scale groundwater monitoring carried out over a period of nine years. The objectives of the work were to define the conceptual hydrogeological and hydrogeochemical model of two aquifers of Ágreda and to identify the origin of the high sulphate content of carbonate and hydrogen sulphide Quaternary aquifers and their relationship to lithology and content of pyrites assessing the importance of pyrite oxidation in aerobic and anaerobic hydrogeological environments.

The manuscript is well written and can represent an important contribution to the scientific literature. However, I believe this manuscript should be published after a minor revision.

Comments (P = page/ R = row):

P3/R99-106: I suggest moving this section in the introduction of the “General characteristics of the area”.

P6/R240-252: I suggest moving all this paragraph in “Distribution of pyrites”.

Figures 2 and 3: Please improve the quality of the figures increasing the size of photos and texts.

P12/R475: The figures in the text should be mentioned in increasing order (Figure 5 before Figure 6).

P12/6.1. Alteration (oxidation) of pyrites: This section seems a bit of a repetition. I suggest bringing it back to the results " Distribution of pyrites". The section could be renamed: Distribution and Alteration (Oxidation) of pyrites.

P13/6.2. Hydrogeochemistry: For a better comprehension of the geochemical groups and the processes in act characterizing the studied area, it could be useful add a TIS diagram and a piper (or triangular) diagram to get an immediate idea of the salinity of each individual source.

See:

Apollaro C., Tripodi V., Vespasiano G., De Rosa R., Dotsika E., Fuoco I., Critelli S., Muto F. (2019). Chemical, isotopic and geotectonic relations of the warm and cold waters of the Galatro and Antonimina thermal areas, southern Calabria, Italy. Marine and Petroleum Geology. 109, 469–483. https://doi.org/10.1016/j.marpetgeo.2019.06.020

Or

Apollaro C., Caracausi A., Paternoster M., Randazzo P., Aiuppa A., De Rosa R., Fuoco I., Mongelli G., Muto F., Vannia E. and Vespasiano G. (2020) Fluid geochemistry in a low-enthalpy geothermal field along a sector of southern Apennines chain (Italy).  Journal of Geochemical Exploration.  https://doi.org/10.1016/j.gexplo.2020.106618

P16/Figure 5: The figure is not well explained in the text. Please, improve the description.

P18/613: Figure 8?

P22/R819: Is not clear the concentration of nitrate

Author Response

Dear reviewer, 

Thank you for your contribution. Please find attached our answer.

Best regards,

Authors.

Reviewer 2 Report

see the comments in thw document. the highlighted sentences need rephrase

Author Response

(The authors gave the same response as above.)
